# Evolving graph attention networks for dynamic link prediction

Yucai Jiang[1], Rongying Shan[2]*, Gang Fu[3], Zhuolin Li[2], Jingjing Sun[4], Zhongyun Bao[2]*, Feifei Wei[5]*

**1** School of Automotive and Mechanical Engineering, Lu'an Vocational Technical College, Lu'an, Anhui, China, **2** School of Computer and Information, Anhui Polytechnic University, Wuhu, Anhui, China, **3** Visual Computing Research Center, Shenzhen University, Shenzhen, Guangdong, China, **4** School of Big Data and Artificial Intelligence, Anhui Institute of Information Technology, Wuhu, Anhui, China, **5** School of Computer and Information, Anhui Normal University, Wuhu, Anhui, China

* rongyingshan@stu.ahpu.edu.cn (RS); zhongyunbao@ahpu.edu.cn (ZB); wffphi@ahnu.edu.cn (FW)

## Abstract

Graph neural networks (GNNs), which learn node representations via aggregating their neighbors, have shown superior performance and become the de facto efficient toolkit for analyzing and learning from data with structured properties. However, most existing GNNs are designed for static graphs and assume fixed graph structures and node sets. In many real-world applications, graphs evolve continuously over time—with nodes and edges appearing or disappearing—rendering static models insufficient for capturing these temporal dynamics. In this paper, we propose Evolving Graph Attention Networks (EGAT), a novel framework for dynamic graph representation learning. Specifically, EGAT leverages the anisotropic attention mechanism of Graph Attention Networks (GATs) to capture complex inter-node relationships. Crucially, the multi-head attention weights of the GAT are evolved over time via a recurrent neural network (RNN), enabling the model to adaptively adjust the importance of different neighbors as the graph topology and relational dynamics change. This weight-evolving paradigm couples the anisotropic attention mechanism of GATs with a recurrent subnetwork, enabling the joint modeling of topological evolution and temporal relational dynamics. Extensive experiments on benchmark datasets demonstrate that the proposed model consistently outperforms state-of-the-art baselines.

## Introduction

Graph Neural Networks (GNNs) have attracted great attention for its excellent ability to model the ubiquitous structured data in the real-world, such as natural language processing [1,2], computer vision [3,4], chemistry [5] and point cloud [6,7]. A graph is comprised of edges and nodes, in which the nodes are used to represent the entities and the edges represent relationships between the entities. For most of the GNNs, they resort to the special message passing mechanism [5], in which the message is

**Data availability statement:** The data involved in this study are all publicly available and accessible. The BC-Alpha Dataset can be obtained at http://snap.stanford.edu/data/soc-sign-bitcoin-alpha.html. The UCI Dataset can be obtained at http://Konect.cc/Networks/opsahl-ucsocial/ The AS Dataset can be obtained at http://snap.stanford.edu/data/as-733.html.

**Funding:** This research was supported by the Outstanding Talents Training Program of Anhui Higher Education Institutions in 2021 (Grant No. gxbjZD2021112), funded by the Department of Education of Anhui Province, China.

**Competing interests:** The authors have declared that no competing interests exist.

flowing along the edge between the nodes for the update of each node in the next layers. To obtain the long-range node dependencies, multiple layers can be stacked to propagate information from multiple-hops [8]. Afterwards, at the final layer, the nodes and edges can be formulated to build the downstream task, for instance node classification and link prediction.

These GNNs are usually built on the static scenarios, where graph structure and the total amount of the nodes are always not changed. However, in the real-world, the scene is evolved with the around environments, which poses the challenges to develop the dynamic model that adapts to the real-life scenarios. For instance, users in the social network make a friend or lose a friend over time, then the node representation should be subsequently updated accordingly. Additionally, in financial domain, transactions involve the time stamps which characterize the nature of the user account, such as money laundering and fraud. From the perspective of these realistic scenarios, it is crucial to construct the dynamically evolving model to adapt to the sophisticated applications.

Recently, although the traditional graph convolutional networks (GCNs) [9] has achieved tremendous success with its simplicity and effectiveness and the graph attention networks (GATs) [10] is equipped with building the multi-relation between the nodes, they are developed for static scenarios, thus being insufficient to model the constantly changing world. In this work, based on the GATs, we build a model that dynamically evolves with temporal dimension by a RNNs to update the parameters for capturing the dynamism of the nodes and edges, which is consistent with the evolving sequence of real-world. As a result, our attention mechanism with evolving graph has the significant advancement compared to the existing GATs when modelling the dynamical world.

Similar works also proposed the method based on the GNNs and RNNs [11–13], where the GNNs is leveraged to extract node feature and the RNNs is utilized for temporal learning combined with the learned node feature. However, at the temporal axis, they only utilize one GNNs to model all the graphs, which necessitates acquire the information of the nodes along the time span, thus hardly being promising the performance when the graph is changed. The subsequent work [14] attempts to incorporate the temporal dimension to the GNNs for modelling the dynamical graph structure. However, owing to its simple vanilla GNNs framework, it can not fully investigate the sophisticated scenarios in the real-world.

To solve the problem of constantly changing graphs that may add new node or construct new connection between nodes, in this paper, we propose an evolving graph attention networks (EGAT) that captures the dynamism of the graph along the time span. Specifically, we evolve the multi-head weights of the GATs along the temporal span by using the RNNs at every time step, which can efficiently adapt the model to the real scenarios. Besides, our model can well tackle the new node that has no historical information.

The contributions of the paper can be summarized as follows:

1. We introduce a novel weight-evolving paradigm coupling a recurrent subnetwork with the anisotropic attention mechanism of Graph Attention Networks. This design

enables the model to jointly capture topological evolution and the fine-grained temporal dynamics of relational strengths between nodes.

2. We propose a novel model EGAT that dynamically adapts to the constantly changing real-world for capturing the evolving graph structure and interplay between nodes, in which the nodes and edges are in the status of changing.

3. The extensive experiments have been conducted on a variety of real-life benchmark datasets. And the results showcase that our performances outperform the state-of-the-art work strikingly in almost every aspects.

## Related work

In this section, the graph neural networks and the closed dynamic graphs are introduced.

**Graph Neural Networks.** The most famous work is the GCNs model [9] that linearly approximates the localized spectral convolution and performs the iteratively update of the node embedding through the isotropic averaging over the neighbour nodes embedding. Then, a lot of follow-up works [10,15–18] are inspired in the spatial domain with improvements at different aspects. Especially, the GATs [10] propose the anisotropy scheme when aggregating the neighbour nodes, which learns the importance over the neighbour nodes feature and thus significantly improves the learning capacity by its multi-head architecture.

**Dynamic Graphs.** Indeed, dynamic graphs are developed to tackle the constantly changing scenarios and often derived from the static graphs, which specifically focus on the temporal dimension and corresponding update methods. Recently, dynamic graphs are becoming emerging and have been broadly investigated in academia and industry [19–21]. For instance, some models leverage regularization over the static graph embedding [22,23]. The work [22] gradually refreshes the embedding through utilizing the incremental Singular Value Decomposition (SVD) and then the SVD is obtained when the error is out of the threshold. Another line of work [24,25] is based on random walk, which obtains the transition probabilities by computing the normalized inner products on the node embedding from the past, thus maximizing the probabilities of the sampled random walks.

With the deep learning surging and achieving remarkable success on a wide range of applications, such as in physics [26], in knowledge graphs [27], there have been work DynGEM [28] that attempts to leverage the auto-encoding framework for minimizing the reconstruction loss and distance between similar nodes, thus resulting in clustering of nodes in the embedding space. The salients of DynGEM are that it can be adapted to the constantly changing size of the graph and the history information can also be learned for further initialization for the next training step.

It is worth to mention that another investigation direction of dynamic graphs is the point processes [29–31]. Specifically, Know-Evolve [29] and DyRep [30] leverage the point process to model the edge and take into the input consideration for parameterizing the intensity function through a neural network. For more sophisticated scenarios, such as triadic closure in which triadic is comprised of three nodes, the work [31] also utilizes the point process to investigate thoroughly from the open form (only two pairs connected) to the closed form (three pairs connected each other).

A recent research direction explicitly models the temporal evolution of GNN parameters. A representative work, EvolveGCN [14], employs an RNN to update the weight matrices of a GCN at each time step, decoupling model capacity from changing graph size. However, EvolveGCN [14] inherits the isotropic nature of GCNs—neighbor information is aggregated uniformly after linear transformation—limiting its ability to capture the temporally-varying importance of different edges. In contrast, our EGAT evolves the anisotropic multi-head attention weights of GATs via an RNN, enabling the model to adaptively re-weight neighbor contributions as the graph evolves. This captures richer dynamic patterns in both structural topology and inter-node relational dynamics.

## Methods

In this section, we make a brief introduction to the Graph Attention Networks (GATs) and weight evolution. With these preliminary knowledge, our model EGAT is demonstrated in the following section.

## Graph attention networks

For GATs, let $G = (V, E)$ be a graph, where $V = \{v_1, v_2, \ldots, v_N\}$ and $E$ denote a set of nodes and edges, respectively. The node embedding $v \in V$ are denoted as $x_v \in \mathbb{R}^F$. Let $X_t = \{x_1, x_2, \ldots, x_N\} \in \mathbb{R}^{N \times F}$ at time $t$. Here, $N$ is denoted as total number of nodes of a graph, which is constantly changing when evolving along the temporal dimension. $F$ represents the feature dimension of a node. Let $A_t \in \mathbb{R}^{N \times N}$ be adjacent matrix of the graph at time $t$, where $A_{i,j} = 1$ if $v_j \in N(v_j)$, otherwise $A_{i,j} = 0$. Similarly, with the scenarios of real-world changing, $A_t$ is also evolved when the node is disappeared or added. The $N(\cdot)$ denotes the neighbor nodes. For the sake of capturing the powerful learning ability of model, GATs leverages the shared linear transformation $w \in \mathbb{R}^{F' \times F}$ for every node, which is followed by a self-attention on the nodes ($a : \mathbb{R}^{F'} \times \mathbb{R}^{F'} \to \mathbb{R}$). Thus, the *att*ention coefficients can be obtained:

$$e_{ij} = a(wx_i, wx_j),$$ (1)

which shows the importance of node $j$ feature representation to node $i$. Then, the softmax function is applied to normalize the coefficients for being comparable across different nodes:

$$\alpha_{ij} = soft\max_j(e_{ij}) = \frac{\exp(e_{ij})}{\sum_{j' \in N_i} \exp(e_{ij'})}.$$ (2)

The aggregated features from the neighbours can be formulated as:

$$x'_i = \sigma \left( \sum_{j \in N_i} \alpha_{ij} w x_j \right).$$ (3)

For multi-head attention, the Eq (3) is independently conducted by $K$ times followed by the concatenating of their features:

$$x'_i = \|_{k=1}^K \sigma \left( \sum_{j \in N_i} \alpha_{ij}^k w^k x_j \right),$$ (4)

where $\|$ denotes the concatenation, $\alpha_{ij}^k$ represents the normalized attention coefficients obtained from the $k$-th attention mechanism $\alpha_{ij}$ and $w^k$ is the $k$-th linear transformation weight matrix corresponding to the $k$-th head.

From the above, it can be seen that the GATs [10] consists of multi-head attention, which is leveraged to model the sophisticated relationship between the nodes.

## Weight evolution

For the GATs with $k$ heads, we exploit the $k$ RNNs: long short-term memory (LSTM [32]) to evolve to weights $w^k$. Note that the LSTM is not the only choice, and other RNNs, such as gated recurrent unit (GRU), is alternative. In this setting, the input and output of the LSTM are the $w^k$ of different GATs without the node embedding involving, thus leading to the system information contained by the LSTM cell. The update process can be derived as:

$$w_t^k = LSTM^k \left( w_{t-1}^k \right),$$ (5)

where the $w_t^k$ is denoted as $k$-th head weight at time $t$. The detailed process is demonstrated in *t*he later subsection.

## Evolving graph attention networks

Based on the weight evolution scheme from Eq 5, the evolving graph attention networks (EGAT) can be derived by evolving the weight of each head. In Fig 1(b), it can be seen that the graph structure evolves from time 0 to time 1 via dynamical multi-relations between neighboring nodes and the central node, driven by the temporal dynamism of the head weights as illustrated in Fig 2. In contrast, the limited expressive power of vanilla GCN, as shown in Fig 1(a), can hardly capture such dynamism.

This distinction stems from a fundamental difference in aggregation paradigms: prior dynamic models such as EvolveGCN [14] rely on isotropic aggregation, where the temporal evolution remains confined to a global feature extractor and is thus agnostic to fine-grained relational dynamics between individual node pairs. In EGAT, however, the evolved weights are coupled with the spatial domain through the attention mechanism—even when the local topology and node features remain unchanged, the drift in weights induces shifts in pairwise attention scores, enabling the model to capture relation decay or interest shift purely through evolving aggregation logic. Obviously, our method can be extended to any temporal span for adapting to arbitrary dynamic scenarios.

Table 2 details the weight evolution process for each attention head, where the standard LSTM formulation is extended from vector-valued to matrix-valued states, allowing the recurrent unit to directly update the GAT weight matrices over time.

## Experiments

In this section, we provide a detailed experiments to demonstrate the effectiveness and efficiency of our model EGAT. Specifically, we conduct the experiments on multiple data sets, tasks, and compared methods. Moreover, to show the full efficiency, different evaluation metrics are compared. Finally, the best validation epoch is reported in each experiment.

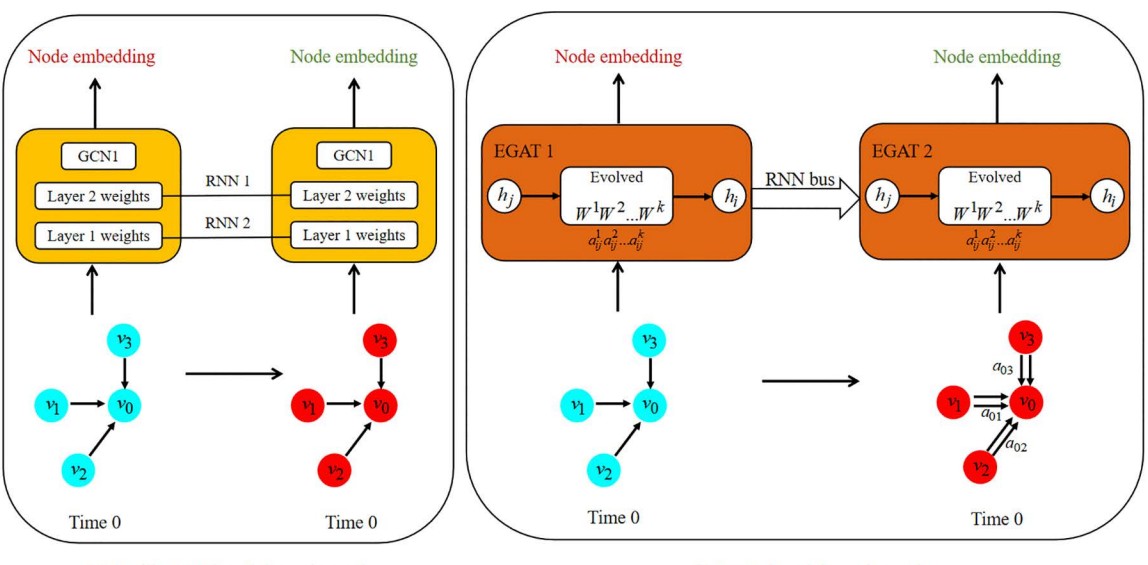

(a) Vanilla GCN based dynamic graph        (b) EGAT based dynamic graph

**Fig 1. (a) The vanilla GCNs based dynamic graph that has limited expressiveness power for sophisticated structure. (b)** The overall structure of our proposed method EGAT. Through the RNN bus, the $k$-th RNN are leveraged to evolve the $k$-th head weight of the GATs at time 0 and 1, in which the multi-relation, namely $\alpha_{ij}$, is evolved between the neighbour nodes to the central node (detailed process can refer to Tables 1 and 2) and then it can expend to more temporal dimensions for adapting to the more sophisticated scenarios.

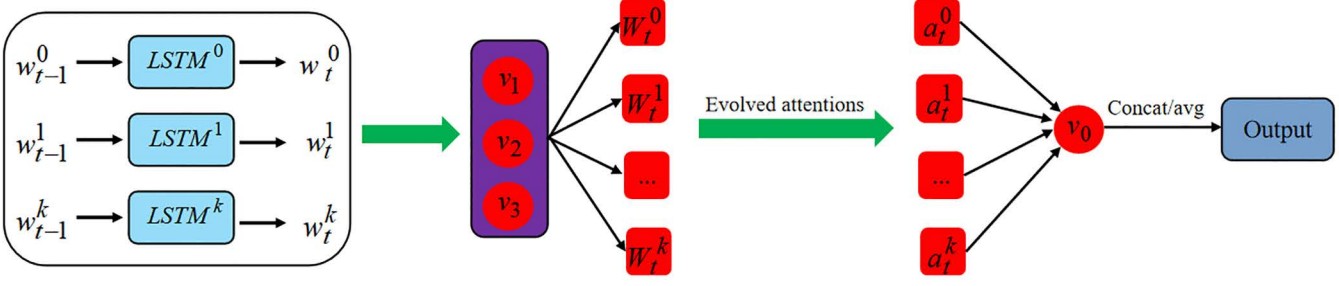

**Fig 2. The illustration of EGAT.** The $v_0$ is denoted as the central nodes, while $v_1$, $v_2$, and $v_3$ are neighbour nodes to $v_0$. The weights $W$ of the $k$ heads are evolved from time $t-1$ to $t$, leading to the *at*tention scores $\alpha$ accordingly, and then these weights are shared with the neighbour nodes to capture the dynamical node embedding and deeply reveal the interplay between the central and neighbour node.

**Table 1. EGAT: Evolve the weight of the $k$ heads at each time step.**

| |
|---|
| **Input:** $A_t, X_t, w_{t-1}^k$. |
| **Output:** $X_1, X_2, \ldots, X_{t+1}$. |
| 1: for $i = 1$ to $k$ do |
| 2: $w_t^k = LSTM^i(w_{t-1}^k)$ |
| 3: $X_{t+1} = GAT(A_t, X_t, w_t^k)$ |
| 4: end for |

**Table 2. Evolving the weights of each head at all time steps.**

| |
|---|
| **Input:** $X_t$. |
| **Output:** $H_t$. |
| 1: $F_t = \text{sigmoid}(w_F X_t + U_F H_{t-1} + B_F)$ |
| 2: $I_t = \text{sigmoid}(w_I X_t + U_I H_{t-1} + B_I)$ |
| 3: $O_t = \text{sigmoid}(w_O X_t + U_O H_{t-1} + B_O)$ |
| 4: $\tilde{C}_t = \tanh(w_C X_t + U_C H_{t-1} + B_C)$ |
| 5: $C_t = F_t \circ C_{t-1} + I_t \circ \tilde{C}_t$ |
| 6: $H_t = O_t \circ \tanh(C_t)$ |

## Datasets description

We conduct the experiments on the most widely publicly used benchmark datasets.

**SBM Dataset.** It is a widely used random graph model that is generated by SBM model. With the in-block and cross-block probability set to be 0.2 and 0.1 respectively, we generate the initial snapshot of the dynamic graph. Next, it randomly generated 10–20 nodes at each time step, which are added to another community. The final synthetic SBM incorporates 1000 nodes, 4 870 863 edges, and 50 timestamps.

**BC-Alpha Dataset.** It is a message communication network in the form of who-trusts-whom, in which the users trade by Bitcoin. The BC-Alpha dataset consists of 3,777 nodes and 24,173 edges across 136 timestamps.

**Autonomous System (AS) Dataset.** The AS is the second real-world dataset which is leveraged for communication network by exchanging traffic signals with peers. The dataset consists of 6,474 nodes ranging from November 8, 1997, to January 2, 2000.

**UCI Dataset**. UCI consists of an online community from the University of California, Irvine, in which the user send messages to each other that indicate the edge between the users. It is comprised of 1,899 nodes and 59,835 edges spanning 88 timestamps (directed graph), which shows a high dynamism in terms of transition state.

The statistics of the datasets and their split scheme of train, validation and test can be seen on Table 3.

## Compared methods

We make a thorough comparison with these baselines. **GCN** [9]. This is a static model that is built without temporal information with single GCN applied to all the time steps.

**GCN-GRU** [33] and **GCN-LSTM** [11]. These two models are deep learning frameworks that integrate Graph Convolutional Networks (GCNs) with recurrent units (GRU/LSTM) to model spatiotemporal dependencies in dynamic graph data, highly effective for tasks requiring the prediction of dynamic node-level opinions or states in large-scale, time-varying networks.

**DynGEM**. [28] It is a dynamic graph embedding model that uses deep autoencoders to generate stable, low-dimensional node representations for evolving graphs. Its key strengths include temporal stability, scalability for growing graphs, and computational efficiency compared to static embedding methods applied per snapshot.

**dyngraph2vec** and **dyngraph2vecAERNN**. [34] This method has high similarity to DynGEM while additionally contain the former node information. It has several variants of dyngraph2vecAE, dyngraph2vecRNN, and dyngraph2vecAERNN.

**EvolveGCN**. [14] It is a dynamic graph neural network model that captures temporal evolution in graph-structured data, which adapts Graph Convolutional Network (GCN) parameters directly over time using a recurrent mechanism, enabling it to handle dynamic graphs with changing node sets and maintains strong performance in tasks such as link prediction, edge classification, and node classification. COMP-GCN. [35] It proposes the graph Fourier transformation in simplex space, based on which a compositional graph convolutional network layer is introduced.

**EGAI**. [36] Its core idea is to dynamically filter out harmful or redundant information from neighbor nodes during training by removing specific edges, thereby improving model performance and mitigating over-smoothing.

**ADMP-GNN**. [37] This framework addresses the limitation of GNNs using a fixed number of message-passing layers for all nodes, allowing each node to dynamically determine its optimal number of propagation steps based on its local characteristics, leading to improved performance on node classification tasks.

$\beta$-**GNN**. [38] This model employs a learned, dynamic weighting mechanism, which forms a weighted ensemble between any base GNN and an MLP. The weight adaptively modulates the GNN's influence, helping to preserve performance on clean data while improving resilience to attacks.

**HGCN**. [39] It proposes a self-tuning toolkit using GCN models, which integrates multiple graph representations of event sequences with different choices of node- and graph-level attributes and in temporal dependencies via edge weights.

**Table 3. Statistics of the datasets.**

| Datasets | Nodes | Edges | TimeSteps (Train/Val/Test) |
|---|---|---|---|
| SBM | 1,000 | 4,870,863 | 35/5/10 |
| BC-Alpha | 3,777 | 24,173 | 95/13/28 |
| AS | 6,474 | 13,895 | 70/10/20 |
| UCI | 1,899 | 59,835 | 62/9/17 |

## 0.1 Results for link prediction and discussion

As shown in Tables 4 and 5, our proposed method demonstrates comprehensive performance advantages in the dynamic graph link prediction task. Across four datasets SBM, BC-Alpha, UCI, and AS, our method achieves the best performance on the core metric MAP. In terms of MRR, our method also outperforms others on the SBM and UCI datasets. All experiments were repeated ten times with different random seeds, reporting the mean and standard deviation of MAP and MRR. Paired t-tests conducted between EGAT and the best-performing baseline on each dataset confirm the improvements achieved by EGAT are statistically significant ($p < 0.05$) for the MAP metric on all four datasets, and for the MRR metric on

**Table 4. Results for link prediction (MAP). Each column represents one dataset. Results are reported as mean ± standard deviation over five independent runs. The best mean results are shown in bold.** $*$ **indicates statistically significant improvement over the best baseline (paired t-test, $p < 0.05$).**

| Model | SBM | BC-Alpha | UCI | AS |
|---|---|---|---|---|
| Static GCN | 0.1978±0.0012 | 0.0003±0.0001 | 0.0247±0.0008 | 0.0003±0.0001 |
| GCN-LSTM | 0.1889±0.0015 | 0.0002±0.0001 | 0.0100±0.0006 | 0.0498±0.0011 |
| GCN-GRU | 0.1890±0.0014 | 0.0001±0.0000 | 0.0111±0.0007 | 0.0707±0.0013 |
| DynGEM | 0.1672±0.0021 | 0.0520±0.0010 | 0.0205±0.0009 | 0.0523±0.0012 |
| Dyngraph2vecAE | 0.0976±0.0018 | 0.0501±0.0012 | 0.0042±0.0003 | 0.0326±0.0010 |
| Dyngraph2vecAERNN | 0.1585±0.0016 | 0.1092±0.0015 | 0.0201±0.0008 | 0.0705±0.0014 |
| EvolveGCN | 0.1981±0.0013 | 0.0035±0.0002 | 0.0266±0.0007 | 0.1131±0.0016 |
| COMP-GCN | 0.2126±0.0015 | 0.0919±0.0014 | 0.0279±0.0008 | 0.1418±0.0017 |
| EGAI | 0.2301±0.0014 | 0.0845±0.0016 | 0.0287±0.0009 | 0.1295±0.0015 |
| ADMP-GNN | 0.2089±0.0015 | 0.1179±0.0013 | 0.0305±0.0007 | 0.1607±0.0014 |
| $\beta$-GNN | 0.2073±0.0016 | 0.1115±0.0015 | 0.0282±0.0008 | 0.1597±0.0016 |
| HGCN | 0.2349±0.0012 | 0.1037±0.0014 | 0.0298±0.0007 | 0.1578±0.0015 |
| **Ours** | **0.2400±0.0010**$^*$ | **0.1202±0.0011**$^*$ | **0.0306±0.0006**$^*$ | **0.1612±0.0012**$^*$ |

**Table 5. Results for link prediction (MRR). Each column represents one dataset. Results are reported as mean ± standard deviation over five independent runs. The best mean results are shown in bold.** $*$ **indicates statistically significant improvement over the best baseline (paired t-test, $p < 0.05$).**

| Model | SBM | BC-Alpha | UCI | AS |
|---|---|---|---|---|
| Static GCN | 0.0135±0.0004 | 0.0030±0.0002 | 0.1134±0.0015 | 0.0550±0.0012 |
| GCN-LSTM | 0.0118±0.0005 | 0.0003±0.0001 | 0.0967±0.0018 | 0.3205±0.0025 |
| GCN-GRU | 0.0116±0.0005 | 0.0004±0.0001 | 0.0979±0.0017 | 0.3376±0.0028 |
| DynGEM | 0.0136±0.0006 | 0.1280±0.0015 | 0.1049±0.0016 | 0.1021±0.0018 |
| Dyngraph2vecAE | 0.0077±0.0004 | 0.1471±0.0018 | 0.0535±0.0012 | 0.0693±0.0015 |
| Dyngraph2vecAERNN | 0.0117±0.0005 | **0.1937±0.0020** | 0.0708±0.0014 | 0.0488±0.0013 |
| EvolveGCN | 0.0135±0.0004 | 0.1178±0.0016 | 0.1372±0.0015 | 0.2738±0.0022 |
| COMP-GCN | 0.0132±0.0005 | 0.1695±0.0017 | 0.1376±0.0014 | **0.2849±0.0021** |
| EGAI | 0.0119±0.0004 | 0.1251±0.0015 | 0.1205±0.0016 | 0.2819±0.0020 |
| ADMP-GNN | 0.0166±0.0005 | 0.1593±0.0016 | 0.1176±0.0015 | 0.2381±0.0019 |
| $\beta$-GNN | 0.0168±0.0005 | 0.1510±0.0017 | 0.1048±0.0016 | 0.2438±0.0020 |
| HGCN | 0.0151±0.0004 | 0.1429±0.0015 | 0.1220±0.0014 | 0.2646±0.0018 |
| **Ours** | **0.0169±0.0003**$^*$ | 0.1313±0.0012 | **0.1414±0.0011**$^*$ | 0.2505±0.0016 |

the SBM and UCI datasets. These results collectively demonstrate both the effectiveness and the stability of our proposed method.

While existing methods focus on different aspects of dynamic modeling, they exhibit notable limitations. For instance, EvolveGCN [14] updates GCN parameters solely through recurrent units, which struggles to capture the complex, global dynamic interactions within the evolving graph structure. ADMP-GNN [37] though implements node-level adaptability in the depth of message passing, remains constrained by using fixed, non-evolving transformation weights during each aggregation step. This limits its ability to adapt the nature of information fusion across time and topology. Methods like EGAI [36] aim to enhance GNNs by dynamically filtering out harmful neighbor information during training, which primarily addresses static graph quality issues rather than temporal evolution. Its edge-dropping strategy does not inherently model the temporal dynamics or the evolution of transformation functions crucial for dynamic link prediction.

In contrast, our method fundamentally advances dynamic graph modeling by evolving the anisotropic attention weights of Graph Attention Networks via a recurrent subnetwork. This design enables two key advantages: first, by evolving multi-head attention mechanisms rather than homogeneous convolution filters, our model captures the temporal dynamics of relational strengths between central nodes and their neighbors—an aspect that the isotropic weight evolution of EvolveGCN [14] cannot express. Second, by jointly evolving the transformation weights with the message-passing topology, our approach not only considers where information flows (as in ADMP-GNN [37]) but also adaptively modulates how information is transformed and integrated at each step, offering a more nuanced mechanism for capturing dynamic dependencies than the static or filtering-based approaches of models like EGAI [36].

## 0.2 Computational efficiency analysis

We further analyze the computational efficiency of EGAT compared to representative dynamic graph models. Table 3 reports the average training time per epoch and parameter counts on the SBM dataset.

**Theoretical Complexity.** For a graph with $|V|$ nodes and $|E|$ edges, let $F$ and $F'$ denote input and output feature dimensions, and $K$ the number of attention heads. Following GAT [10], a single head requires $O(|V|FF' + |E|F')$ operations. EGAT extends this with LSTM-based weight evolution, which incurs $O(K \cdot (FF')^2)$ per layer for evolving $K$ independent transformation matrices. The overall per-time-step complexity is:

$$O\left(K \cdot (|V|FF' + |E|F') + K \cdot (FF')^2\right).$$

(6)

For comparison, we examine three representative baselines with distinct architectural choices:

**Static GAT [10]:** As established in the original work, the per-time-step complexity for $K$ attention heads is $O(K \cdot (|V|FF' + |E|F'))$, since the individual heads' computations are fully independent and can be parallelized. No weight evolution overhead is incurred, but this parameter sharing fundamentally limits the model's capacity to adapt to temporal distribution shifts.

**EvolveGCN [14]:** This model evolves the weight matrix of a standard GCN layer through an LSTM. The per-time-step cost consists of the GCN convolution operation and the LSTM-based weight update. The GCN convolution exhibits complexity similar to a single attention head without the $K$ factor (i.e., isotropic aggregation). The LSTM update cost scales with the square of the weight matrix size, as the recurrent cell operates on the flattened parameter vector. Critically, EvolveGCN does not employ multi-head attention, resulting in a constant-factor reduction compared to EGAT.

**EGAT (Ours):** Our method integrates multi-head attention with temporally evolving weights. Relative to EvolveGCN, EGAT introduces an additional factor of $K$ in both the graph convolution and weight evolution terms. Relative to static GAT, EGAT adds the $O(K \cdot (FF')^2)$ weight evolution overhead. Importantly, this additional term is independent of the graph scale ($|V|$ and $|E|$) and thus constitutes a negligible fraction of total runtime on large graphs. Furthermore, as noted in [10], the $K$ attention heads are independent and can be computed in parallel.

**Efficiency Summary.** As reported in Table 6, EGAT requires 1.25× the training time of EvolveGCN and 1.11× that of static GAT, while using only 48,156 parameters—a modest 6% increase over EvolveGCN (45,320) and 7% over static GAT (44,892). The observed runtime ratio is substantially below the theoretical $K=8$ factor because the weight evolution component accounts for <5% of total computation and multi-head operations are efficiently GPU-parallelized [10]. Parameter efficiency stems from the $K$ heads sharing the same LSTM evolution mechanism, with additional parameters limited to per-head attention coefficients that scale as $O(K \cdot F')$ rather than $O(K \cdot FF')$. Since dynamic graph models are typically trained offline, this modest overhead is well within acceptable limits for practical deployment.

### 0.3  Interpretability analysis of evolving transformation weights

As shown in Fig 3, each curve depicts the Frobenius norm of the linear transformation matrix $W_k$ for the $k$-th attention head in EGAT. The distinct temporal patterns across heads demonstrate that our model dynamically updates transformation weights. Head 1 shows a strong rising trend, capturing persistent structural evolution, while Heads 5 and 8 exhibit steady declines. The diverse evolving patterns of weights validate that EGAT adaptively modulates feature transformations to match varying topological dynamics, significantly improving the interpretability of dynamic graph representation learning.

**Table 6.  Efficiency comparison on SBM.**

| Model | Time/Epoch (s) | Parameters | Rel. Time |
|---|---|---|---|
| Static GAT [10] | 3.34 | 44,892 | 1.00× |
| EvolveGCN [14] | 2.96 | 45,320 | 0.88× |
| **EGAT (Ours)** | **3.71** | **48,156** | **1.11×** |

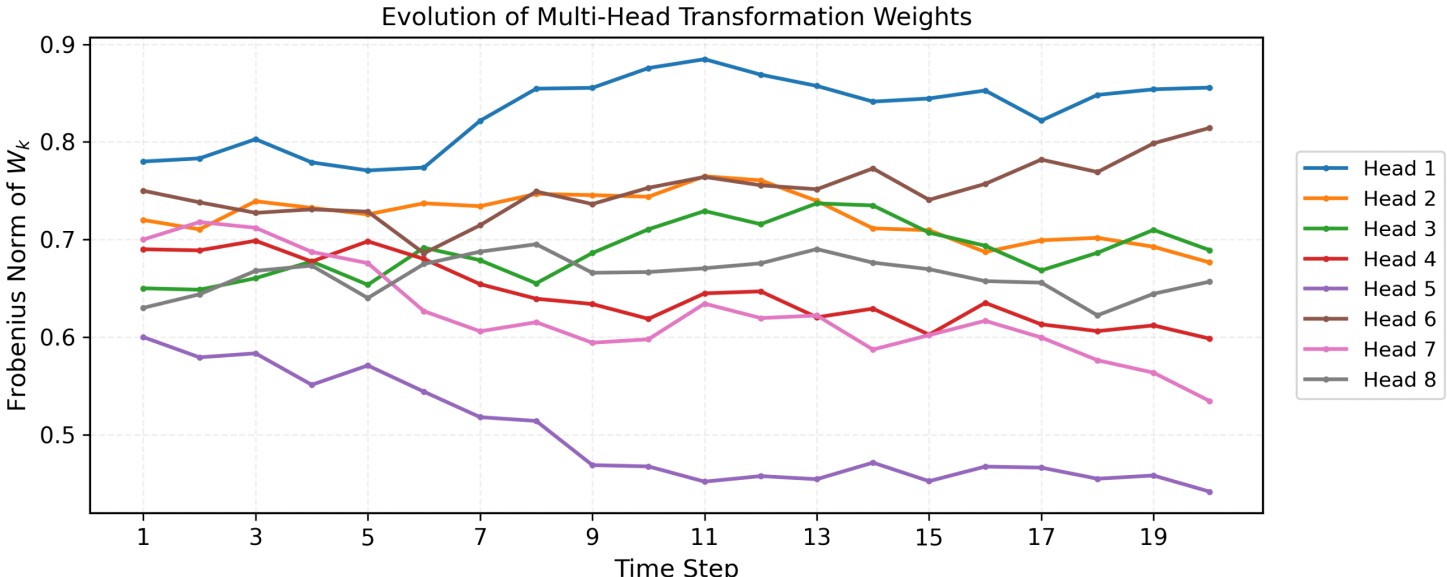

**Fig 3.  Evolution of multi-head transformation weights over time.**

## Conclusions and future works

In this paper, we proposed EGAT, a dynamic graph neural network that evolves the multi-head attention weights of Graph Attention Networks via a recurrent neural network. By coupling RNN-based weight evolution with anisotropic neighborhood aggregation, EGAT effectively captures both the structural evolution of the graph and the temporal variation in relational strengths between nodes. Extensive experiments on four benchmark datasets demonstrate that EGAT consistently outperforms state-of-the-art dynamic graph models on link prediction tasks, establishing a strong new baseline for dynamic graph representation learning.

**Future Works.** The EGAT framework opens several promising avenues for further investigation. First, the weight evolution paradigm presented in this work is naturally extensible to more advanced sequence modeling backbones. In particular, replacing the recurrent unit with a Transformer-based temporal encoder [40] could allow the model to explicitly attend to long-range historical weight configurations via self-attention, offering an alternative mechanism for capturing cross-time dependencies. Preliminary exploration of this Transformer-enhanced variant constitutes an interesting direction for future research. Second, investigating the spectral properties of evolving attention weights—specifically how the graph Fourier basis transforms [41] over time —may yield deeper theoretical insights into the behavior of dynamic attention mechanisms. Finally, extending EGAT to continuous-time dynamic graphs and evaluating its scalability on industrial-scale datasets represent important practical next steps.

## Author contributions

**Conceptualization:** Yucai Jiang, Zhongyun Bao, Feifei Wei.

**Data curation:** Yucai Jiang, Gang Fu, Zhuolin Li, Jingjing Sun, Zhongyun Bao.

**Formal analysis:** Yucai Jiang, Gang Fu, Zhuolin Li, Jingjing Sun, Zhongyun Bao, Feifei Wei.

**Investigation:** Yucai Jiang, Rongying Shan.

**Visualization:** Gang Fu, Zhuolin Li.

**Writing – original draft:** Yucai Jiang, Rongying Shan.

**Writing – review & editing:** Gang Fu, Zhuolin Li, Jingjing Sun, Zhongyun Bao, Feifei Wei.

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
