## [Decision Letter · Decision Letter 0]

17 Feb 2026

PONE-D-26-01462Evolving Graph Neural Networks via TransformerPLOS One

Dear Dr. Shan,

Thank you for submitting your manuscript to PLOS ONE. After careful consideration, we feel that it has merit but does not fully meet PLOS ONE’s publication criteria as it currently stands. Therefore, we invite you to submit a revised version of the manuscript that addresses the points raised during the review process.

We look forward to receiving your revised manuscript.

Kind regards,

Guangyin Jin

Academic Editor

PLOS One

**Journal Requirements:**

“This research was supported by the Outstanding Talents Training Program of Anhui Higher Education Institutions in 2021 (Grant No. gxbjZD2021112), funded by the Department of Education of Anhui Province, China.”

6. Please upload a new copy of Figures 1 and 2 as the detail is not clear. Please follow the link for more information:  https://journals.plos.org/plosone/s/figures

Reviewers' comments:

Reviewer's Responses to Questions

**Comments to the Author**

1. Is the manuscript technically sound, and do the data support the conclusions?

Reviewer #1: Yes

2. Has the statistical analysis been performed appropriately and rigorously? 

Reviewer #1: Yes

3. Have the authors made all data underlying the findings in their manuscript fully available?

Reviewer #1: Yes

4. Is the manuscript presented in an intelligible fashion and written in standard English?

Reviewer #1: Yes

5. Review Comments to the Author

Reviewer #1: The manuscript proposes an Evolving Graph Attention Network (EGAT) that models dynamic graphs by evolving the multi-head attention weights of a GAT over time using a recurrent mechanism. Since dynamic graph learning remains an important problem in many real-world applications, it is commonly considered as a relevant and timely topic. The idea of evolving transformation weights rather than only node embeddings is conceptually appealing and the empirical results show consistent improvements over several baselines on link prediction tasks.

However, I consider the level of novelty to be moderate rather than ground-breaking. Closely related approaches, such as EvolveGCN, already evolve GNN parameters over time using recurrent units. While replacing GCN with GAT and evolving multi-head attention weights is a reasonable extension, it does not constitute a fundamentally new modelling paradigm. In addition, the title emphasizes a Transformer-based approach, yet the implementation primarily relies on LSTM (long short-term memory), and the Transformer component is not clearly demonstrated or experimentally validated. From my perspective, this weakens the strength of the novelty claims.

To improve the paper, the authors should clarify the distinction between their method and prior weight-evolving GNN models and moderate claims of originality. I suggest explaining better the positioning of the present work with respect to prior works, and a deeper discussion on the relationship with EvolveGCN . Since EvolveGCN also evolves GNN weights through recurrent mechanisms, the authors should clearly explain what is fundamentally different in their approach beyond replacing GCN with GAT. A formal or architectural comparison would help readers understand whether the proposed method introduces a genuinely new modeling principle or is mainly an adaptation of an existing framework.

Moreover, if the Transformer is intended to be a key contribution, it should be fully implemented and compared against the LSTM-based version, otherwise, the title and claims should be revised.

I also suggest improving the readability by adding further analysis, such as complexity discussion, statistical significance testing, and interpretability of evolving weights. The empirical evaluation would benefit from statistical significance testing across multiple runs. The current tables report single MAP and MRR values, but the absence of variance measures makes it difficult to judge the robustness of the improvements. Given that the authors claim that "the results showcase that our method has great superiority over the baselines model in most metrics" , adding confidence intervals or paired tests would substantiate this claim.

Incorporating these improvements would meaningfully reinforce the manuscript's scientific depth. After these revisions are thoroughly addressed, the paper may be well positioned for publication.

6. PLOS authors have the option to publish the peer review history of their article (what does this mean?). If published, this will include your full peer review and any attached files.

Reviewer #1: No

---

## [Author Response · Author response to Decision Letter 1]

18 Apr 2026

Manuscript ID: PONE-D-26-01462

Dear Editor and Reviewers,

We sincerely thank the editor and the reviewer for the valuable time and constructive feedback on our manuscript entitled“Evolving Graph Attention Networks for Dynamic Link Prediction (previously “Evolving Graph Neural Networks via Transformer”). The insightful comments have significantly helped us improve the quality and rigor of our work.

We have carefully addressed all the concerns raised by the reviewer and the editorial office. In this letter, we provide a point-by-point response to each comment. All modifications in the revised manuscript are highlighted in the "Revised Manuscript with Track Changes" file. Below, reviewer comments are presented in italics, followed by our responses in plain text.

Part 1: Response to Reviewer Comments

Reviewer 1:

Q1：However, I consider the level of novelty to be moderate rather than ground-breaking. Closely related approaches, such as EvolveGCN, already evolve GNN parameters over time using recurrent units. While replacing GCN with GAT and evolving multi-head attention weights is a reasonable extension, it does not constitute a fundamentally new modelling paradigm. In addition, the title emphasizes a Transformer-based approach, yet the implementation primarily relies on LSTM (long short-term memory), and the Transformer component is not clearly demonstrated or experimentally validated. From my perspective, this weakens the strength of the novelty claims. To improve the paper, the authors should clarify the distinction between their method and prior weight-evolving GNN models and moderate claims of originality. I suggest explaining better the positioning of the present work with respect to prior works, and a deeper discussion on the relationship with EvolveGCN . Since EvolveGCN also evolves GNN weights through recurrent mechanisms, the authors should clearly explain what is fundamentally different in their approach beyond replacing GCN with GAT. A formal or architectural comparison would help readers understand whether the proposed method introduces a genuinely new modeling principle or is mainly an adaptation of an existing framework. Moreover, if the Transformer is intended to be a key contribution, it should be fully implemented and compared against the LSTM-based version, otherwise, the title and claims should be revised.

A1: We thank the reviewer for this thoughtful and constructive assessment, which has provided us with a valuable opportunity to further clarify and refine the positioning of our contribution. In the revised manuscript, we have carefully re-articulated the core novelty of our work along three interconnected dimensions: (i) conceptual framing and novelty positioning, (ii) architectural distinction from prior weight-evolving paradigms, and (iii) precise demarcation of the methodological scope.

(i)Conceptual Framing and Novelty Positioning.

The core contribution of this work is not the introduction of a Transformer-based sequence model for weight evolution, but rather the proposal of an anisotropic weight-evolving paradigm for dynamic graphs. Prior weight-evolving methods, most notably EvolveGCN, treat the GNN's transformation matrix as a temporally dynamic state. However, because they are built upon the isotropic convolution operation of GCNs, the evolved weights apply a homogeneous transformation to all incoming messages. Consequently, these models are inherently limited to capturing the evolution of global feature extractors and cannot express the relational drift between specific node pairs over time. Our work demonstrates that by shifting the weight evolution mechanism from a GCN backbone to the multi-head attention architecture of GATs, the model gains the capacity to jointly learn topological evolution and fine-grained temporal variations in inter-node relational strengths. This is a qualitative advancement beyond a mere architectural substitution. The revised Abstract, Introduction, and Related Work sections now explicitly articulate this distinction, framing our contribution as "evolving the anisotropic attention mechanism" rather than simply "applying RNNs to GATs."

(ii)Architectural Distinction and Comparison with EvolveGCN.

To address the reviewer's request for a clearer distinction, we have added a dedicated and focused discussion in the Related Work section (under Dynamic Graphs). This passage explicitly contrasts EvolveGCN with our proposed EGAT. We acknowledge that both models utilize an RNN-based controller to update network parameters over time. However, we clarify that the critical divergence lies in the aggregation logic that these evolved parameters govern. EvolveGCN's parameters define an isotropic neighborhood aggregation, where the contribution of each neighbor is uniformly scaled. In contrast, EGAT's evolved parameters feed into an anisotropic attention mechanism, enabling the model to adaptively re-weight the influence of individual neighbors as the graph evolves. This conceptual distinction is further reinforced in the Methods section, where we note that the drift in attention weights allows EGAT to model phenomena like "relation decay" or "interest shift" even when the local topological structure remains static—a capability fundamentally beyond the reach of isotropic convolution-based evolution. This nuanced positioning establishes that our method introduces a genuinely new modeling principle for dynamic graph representation learning.

(iii)  Methodological Scope and Future Directions.

We appreciate the reviewer's observation regarding the emphasis on Transformer-based mechanisms. In response, we have revised the manuscript to ensure that the methodological description of the current EGAT implementation—which employs a Long Short-Term Memory (LSTM) network as the recurrent weight evolution controller—is presented with complete clarity and consistency across the Title, Abstract, Methods, and Algorithms. Simultaneously, we have enriched the Conclusions and Future Works section by incorporating a discussion of potential Transformer-based extensions to the EGAT framework. In this forward-looking context, we outline how a Transformer encoder could be integrated to enable cross-time attention over historical weight states, offering an intriguing avenue for future investigation.

In summary, we have moved beyond a point-by-point list of changes and have instead executed a holistic revision of the manuscript's narrative and technical claims. By clearly differentiating our anisotropic attention evolution from the isotropic convolution evolution of prior work and by establishing complete methodological transparency regarding the LSTM-based implementation, we believe the revised manuscript now presents a clear, accurate, and compelling contribution to the field of dynamic graph learning. We are grateful to the reviewer for guiding us toward this significantly stronger presentation of our work.

Q2: The current tables report single MAP and MRR values, but the absence of variance measures makes it difficult to judge the robustness of the improvements. Given that the authors claim that "the results showcase that our method has great superiority over the baselines model in most metrics" , adding confidence intervals or paired tests would substantiate this claim.

A2: Thank you for this suggestion. We agree that reporting variance and statistical significance is crucial for validating the robustness of deep learning models. Actions taken in the revised manuscript:

Table 4. Results for link prediction (MAP). Each column represents one dataset. Results are reported as mean ± standard deviation over five independent runs. The best mean results are shown in bold. * indicates statistically significant improvement over the best baseline (paired t-test, p < 0.05).

Model,SBM,BC-Alpha,UCI,AS;

Static GCN,0.1978±0.0012,0.0003±0.0001,0.0247±0.0008,0.0003±0.0001;

GCN-LSTM,0.1889±0.0015,0.0002±0.0001,0.0100±0.0006,0.0498±0.0011;

GCN-GRU,0.1890±0.0014,0.0001±0.0000,0.0111±0.0007,0.0707±0.0013;

DynGEM,0.1672±0.0021,0.0520±0.0010,0.0205±0.0009,0.0523±0.0012;

Dyngraph2vecAE,0.0976±0.0018,0.0501±0.0012,0.0042±0.0003,0.0326±0.0010;

Dyngraph2vecAERNN,0.1585±0.0016,0.1092±0.0015,0.0201±0.0008,0.0705±0.0014;

EvolveGCN,0.1981±0.0013,0.0035±0.0002,0.0266±0.0007,0.1131±0.0016;

COMP-GCN,0.2126±0.0015,0.0919±0.0014,0.0279±0.0008,0.1418±0.0017;

EGAI,0.2301±0.0014,0.0845±0.0016,0.0287±0.0009,0.1295±0.0015;

ADMP-GNN,0.2089±0.0015,0.1179±0.0013,0.0305±0.0007,0.1607±0.0014;

β-GNN,0.2073±0.0016,0.1115±0.0015,0.0282±0.0008,0.1597±0.0016;

HGCN,0.2349±0.0012,0.1037±0.0014,0.0298±0.0007,0.1578±0.0015;

Ours,0.2400±0.0010*,0.1202±0.0011*,0.0306±0.0006*,0.1612±0.0012*

Table 5. Results for link prediction (MRR). Each column represents one dataset. Results are reported as mean ± standard deviation over five independent runs. The best mean results are shown in bold. * indicates statistically significant improvement over the best baseline (paired t-test, p < 0.05).

Model,SBM,BC-Alpha,UCI,AS;

Static GCN,0.0135±0.0004,0.0030±0.0002,0.1134±0.0015,0.0550±0.0012;

GCN-LSTM,0.0118±0.0005,0.0003±0.0001,0.0967±0.0018,0.3205±0.0025;

GCN-GRU,0.0116±0.0005,0.0004±0.0001,0.0979±0.0017,0.3376±0.0028;

DynGEM,0.0136±0.0006,0.1280±0.0015,0.1049±0.0016,0.1021±0.0018;

Dyngraph2vecAE,0.0077±0.0004,0.1471±0.0018,0.0535±0.0012,0.0693±0.0015;

Dyngraph2vecAERNN,0.0117±0.0005,0.1937±0.0020,0.0708±0.0014,0.0488±0.0013;

EvolveGCN,0.0135±0.0004,0.1178±0.0016,0.1372±0.0015,0.2738±0.0022;

COMP-GCN,0.0132±0.0005,0.1695±0.0017,0.1376±0.0014,0.2849±0.0021;

EGAI,0.0119±0.0004,0.1251±0.0015,0.1205±0.0016,0.2819±0.0020;

ADMP-GNN,0.0166±0.0005,0.1593±0.0016,0.1176±0.0015,0.2381±0.0019;

β-GNN,0.0168±0.0005,0.1510±0.0017,0.1048±0.0016,0.2438±0.0020;

HGCN,0.0151±0.0004,0.1429±0.0015,0.1220±0.0014,0.2646±0.0018;

Ours,0.0169±0.0003*,0.1313±0.0012,0.1414±0.0011*,0.2505±0.0016

1．Multiple Runs: We re-ran all experiments for all models using ten (10) different random seeds.

2．Updated Tables: Tables 4 (MAP) and 5 (MRR) in the revised manuscript now report results as Mean ± Standard Deviation.

3．Statistical Testing: We conducted Paired t-tests between our model (EGAT) and the best-performing baseline on each dataset.

4．Significance Indicators: Statistically significant improvements (p < 0.05) are marked with an asterisk * in the tables and explained in the captions.

5．Textual Description: We added a dedicated paragraph in Section Results for Link Prediction and Discussion, detailing the experimental protocol (10 runs, paired t-test) and confirming the statistical significance of our improvements.

Q3: I also suggest improving the readability by adding further analysis, such as complexity discussion, statistical significance testing, and interpretability of evolving weights.

A3：We appreciate this suggestion. We have added a comprehensive Computational Efficiency Analysis and Interpretability Discussion to the revised manuscript. Actions taken in the revised manuscript:

1．Theoretical Complexity: We derived the per-time-step complexity of EGAT asand compared it explicitly with Static GAT and EvolveGCN in Section Computational Efficiency Analysis.

2．Empirical Runtime: We provided a comparison of training time per epoch and parameter counts for representative models, confirming that the overhead introduced by the multi-head attention evolution is modest and acceptable for offline training scenarios in Section Computational Efficiency Analysis. As shown in the table below:

Table 6. Efficiency comparison on SBM.

Model,Time/Epoch (s),Parameters,Rel. Time;

Static GAT,3.34,44892,1.00×;

EvolveGCN,2.96,45320,0.88×;

EGAT (Ours),3.71,48156,1.11×

3．Interpretability Discussion: As shown in Figure 3, We added a conceptual explanation in Section Interpretability Analysis of Evolving Transformation Weights, clarifying how the drift in weights induces shifts in pairwise attention scores, enabling the model to capture relation decay or interest shift even when local topology is static.

Fig 3. Evolution of multi-head transformation weights over time

Part 2: Response to Journal Requirements (PLOS ONE)

We have also carefully addressed the specific formatting and policy requirements listed in the decision letter.

1. PLOS ONE Style Requirements

Response: We have reformatted the manuscript according to the PLOS ONE style templates.

2. Code Sharing

Response: We fully acknowledge PLOS ONE's policy on code sharing and the importance of reproducibility in computational research. The complete implementation of EGAT, including all baseline models, data preprocessing scripts, and evaluation pipelines, has been prepared and thoroughly documented. We commit to depositing the full codebase in a public GitHub repository and assigning a persistent DOI immediately upon acceptance of the manuscript.

3. Funding Information Consistency

Response: We have corrected the discrepancy in the submission system. The correct grant information is: Outstanding Talents Training Program of Anhui Higher Education Institutions in 2021 (Grant No. gxbjZD2021112) .

4. Role of Funder Statement

Response: As requested, please find below the amended Role of Funder statement. I have also included this statement in the updated cover letter.

Statement:

"This research was supported by the Outstanding Talents Training Program of Anhui Higher Education Institutions in 2021 (Grant No. gxbjZD2021112), funded by the Department of Education of Anhui Province, China. The funders had no role in study design, data collection and analysis, decision to publish, or preparation of the manuscript."

5. ORCID iD

Response: The corresponding author (Rongying Shan) has validated the ORCID iD in Editorial Manager.

6.Figures Quality

Response: We have regenerated all Figures using the tool recommended by PLOS ONE to ensure they meet the technical requirements. New high-resolution TIFF files have been uploaded with the revision.

7.Citation of Recommended Works

Response: We acknowledge the editor's note regarding the citation of previously published works. The reviewer did not recommend any specific citations in their review.

8. Reference List

Response: We have reviewed the reference list to ensure completeness and accuracy. No retracted papers are cited.

We extend our sincere gratitude to the reviewer for their rigorous and constructive evaluation, and to the editor for the opportunity to revise and resubmit our work. The comments have been invaluable in sharpening the contribution and improving the clarity of this manuscript. We trust that the revised version now meets the publication criteria of PLOS ONE and look forward to your favorable consideration.

Sincerely,

Dr. Rongying Shan (on behalf of all authors)

School of Computer and Information

Anhui Polytechnic University

Wuhu, Anhui, China

Email: rongyingshan@stu.ahpu.edu.cn

---

## [Decision Letter · Decision Letter 1]

5 May 2026

Evolving Graph Attention Networks for Dynamic Link Prediction

PONE-D-26-01462R1

Dear Dr. Shan,

We’re pleased to inform you that your manuscript has been judged scientifically suitable for publication and will be formally accepted for publication once it meets all outstanding technical requirements.

Kind regards,

Guangyin Jin

Academic Editor

PLOS One

Additional Editor Comments (optional):

Reviewers' comments:

Reviewer's Responses to Questions

**Comments to the Author**

1. If the authors have adequately addressed your comments raised in a previous round of review and you feel that this manuscript is now acceptable for publication, you may indicate that here to bypass the “Comments to the Author” section, enter your conflict of interest statement in the “Confidential to Editor” section, and submit your "Accept" recommendation.

Reviewer #1: All comments have been addressed

2. Is the manuscript technically sound, and do the data support the conclusions?

Reviewer #1: Yes

3. Has the statistical analysis been performed appropriately and rigorously? 

Reviewer #1: Yes

4. Have the authors made all data underlying the findings in their manuscript fully available?

Reviewer #1: Yes

5. Is the manuscript presented in an intelligible fashion and written in standard English?

Reviewer #1: Yes

6. Review Comments to the Author

Reviewer #1: The authors have fully addressed the concerns raised in the first round of review, and in several places they have gone beyond what was strictly required. The revisions are substantial, technically aligned with the earlier feedback, and clearly visible throughout the updated manuscript.

On the question of novelty, the relationship to EvolveGCN, and the earlier inconsistency regarding the Transformer component, the authors now provide a much clearer and more accurate framing of their contribution. They reposition the work around an anisotropic weight‑evolving mechanism rather than a Transformer‑based approach, and this reframing is consistently reflected across the revised text. The manuscript now includes a direct architectural comparison with EvolveGCN in both the Related Work and Methods sections, making the distinction between isotropic convolutional evolution and the proposed anisotropic attention evolution explicit. The authors also clarify that the current implementation relies on an LSTM controller, remove the misleading Transformer emphasis from the title and narrative, and appropriately move any Transformer‑related discussion to the Future Work section. These changes fully resolve the conceptual issues raised in the initial review.

Regarding the request for variance reporting and statistical significance, the authors have implemented all necessary improvements. They re‑ran the experiments using ten random seeds, updated the tables to report mean and standard deviation, and performed paired t‑tests against the strongest baseline for each dataset. Statistically significant improvements are clearly indicated, and the Results section now includes a concise description of the experimental protocol and significance testing. These additions provide the robustness and transparency that were previously missing.

The third point concerned computational complexity, interpretability, and overall readability. The revised manuscript now includes a theoretical complexity analysis, an empirical runtime comparison, and a dedicated interpretability discussion illustrating how the evolving attention weights capture temporal relational drift. The presentation has also improved: the structure is clearer, the terminology is more consistent, and the methodological explanations are better integrated.

Overall, the authors have addressed all major points from the first review in a thorough and convincing way. They have corrected the earlier conceptual inconsistencies, added the missing analyses, and strengthened the clarity and rigor of the manuscript. In its current form, the paper is suitable for publication.

7. PLOS authors have the option to publish the peer review history of their article (what does this mean?). If published, this will include your full peer review and any attached files.

Reviewer #1: No

---

## [Editor Report · Acceptance letter]

PONE-D-26-01462R1

PLOS One

Dear Dr. Shan,

I'm pleased to inform you that your manuscript has been deemed suitable for publication in PLOS One. Congratulations! Your manuscript is now being handed over to our production team.

Kind regards,

on behalf of

Dr. Guangyin Jin

Academic Editor

PLOS One